# Sub-Muscular Direct-to-Implant Immediate Breast Reconstruction in Previously Irradiated Patients Avoiding the Use of ADM: A Preliminary Study

**DOI:** 10.3390/jcm11195856

**Published:** 2022-10-03

**Authors:** Lucrezia Pacchioni, Gianluca Sapino, Irene Laura Lusetti, Giovanna Zaccaria, Pietro G. Di Summa, Giorgio De Santis

**Affiliations:** 1Department of Plastic and Reconstructive Surgery, University Hospital of Modena, 41125 Modena, Italy; 2Department of Plastic, Reconstructive and Hand Surgery, University Hospital of Lausanne, 1011 Lausanne, Switzerland

**Keywords:** breast reconstruction, radiotherapy, breast implant, direct to implant

## Abstract

Background: The aim of this paper is to present a preliminary experience of sub-muscular primary direct-to-implant (DTI) breast reconstruction without acellular dermal matrix (ADM), after salvage mastectomy for local recurrence following prior irradiation. Methods: A retrospective investigation was performed on a prospectively maintained database of breast reconstruction cases at our institution between January 2015 and December 2020. We considered only immediate DTI breast reconstructions without ADM following radiotherapy and salvage mastectomy for local recurrence, with at least a 12-month follow-up. Results: The study considered 18 female patients with an average of 68 years. According to the BREAST-Q questionnaire, all patients reported high levels of “satisfaction with outcome” with good “psychosocial wellness” and “physical impact” related to the reconstruction. The aesthetic evaluation showed a significant difference between the VAS score gave by the patient (mean 6.9) and the surgeon (mean 5.4). No implant exposure occurred in this series. In terms of complications, four patients (22%) suffered from wound dehiscence and were managed conservatively. Three patients (17%) required primary closure in day surgery following superficial mastectomy flap necrosis. Late capsular contracture was seen in seven patients (four Baker stage II and three Baker stage III, totally 39%); however, no patient was willing to undergo implant exchange. Conclusions: DTI breast reconstruction following prior irradiation can be considered as an option in patients who are not good candidates for autologous breast reconstruction. Our general outcomes compared favorably with literature data regarding the use of staged procedures, with acceptable complication rates and levels of patient satisfaction.

## 1. Introduction

Breast cancer represents the most common cancer occurring among women, and it is estimated that at least one in eight women will develop it during their lifetime [1]. As survival rates and life expectations following breast cancer have continued to improve over recent years [2], breast reconstruction has proven to be a critical step in treatment processes, providing significant gains in terms of quality of life, both psycho-physically and sexually, when compared to non-reconstructed patients [3].

Thanks to surgical and medical advancements, treatment strategies for breast cancer patients are in continuous and progressive evolution, and therefore the indications for oncoplastic procedures and adjuvant therapies (e.g., radiotherapy) are increasing compared with radical procedures [4].

Although oncoplastic breast conserving surgery (BCS) is considered safe [5], rates of local recurrence (up to 10–15%) must be acknowledged and their management can be complex [6,7]. In such cases, a salvage mastectomy is often required, and the reconstruction may be challenging due to prior radiation and a subsequent increased rate of complications [8,9]. Autologous breast reconstruction is the procedure of choice for previously irradiated patients [10]. However, it can be surgically challenging and is not always applicable, depending on patients’ comorbidities or wishes. Indeed, despite a reported higher rate of complications and slightly inferior aesthetic results, implant-based breast reconstruction (IBR), increasingly associated with the use of an acellular dermal matrix (ADM) [11], can be an attractive alternative for patients who are not good candidates for flap reconstruction. However, it should be considered that in a post-radiotherapy setting, the use of ADM is debated and there is no consensus about the safety of its use.

Even though the literature has described the use of the tissue expander and prosthesis technique following salvage mastectomy and previous radiotherapy [12], we could not find any report concerning the use of the direct-to-implant (DTI) technique applied in these rare cases.

The aim of this paper is to present, for the first time, a preliminary experience with sub-muscular primary direct-to-implant breast reconstruction, without the use of any ADM, after salvage mastectomy for local recurrence following prior BCS and radiation.

In this study, we analyzed early and late complications as well as aesthetic outcomes and the satisfaction of patients in the long term, in order to provide quantitative data to critically evaluate this technique.

## 2. Patients and Methods

A retrospective investigation was performed on a prospectively maintained database of breast reconstruction cases at our institution between January 2015 and December 2020. A total of 546 breast reconstructions were screened. For this study, we considered only direct-to-implant immediate breast reconstructions following radiotherapy and salvage mastectomy for local recurrence, with at least 12-month follow-up. Patients’ data, comorbidities, and surgical-related details were collected from clinical, operative and anesthesiologic charts. Early (i.e. first-month) and late complications were recorded. The BREAST-Q version 2.0 questionnaire (out of 100 points, with higher scores reflecting better outcomes) was administrated to all patients at their last follow-up appointment, by a resident plastic surgeon not involved in the study [13]. The aesthetic outcome was evaluated both by the patient and the examining clinician, using the visual analogue score (VAS, 0–10). 

Informed consent was obtained from all patients, including approval for photographic and video documentation.

### 2.1. Surgical Technique

Prophylactic antibiotic (cefazolin or clindamycin) was administered at appropriate dosage 30 min before the skin incision. Mastectomy (either skin or nipple sparing) was performed by a general surgeon, with the patient in a supine position with the arm abducted at 90°. All reconstructive procedures were performed or supervised by the senior author (L.P.). When the lateral border of the pectoralis major muscle had been identified, a sub-muscular pocket was created. Mid-sternal attachments (3th to 5th ribs level) of the pectoralis muscle were weakened using a bipolar technique, avoiding complete interruption of the fibers. Lower sternal fibers (at the 6th rib level) were instead sectioned. Leaving the anterior side of the pectoralis major muscle attached to the subcutaneous tissue at the level of the infra-mammary fold (IMF), which was carefully preserved by the general surgeon during the demolitive procedure, the muscle was detached at its abdominal origin from the rectus and obliquus externus sheet, and the pocket under the IMF was deepened by 2–3 cm. A dual plane pocket was therefore created, with the implant covered by the abdominal subcutaneous tissue in the lower portion. A portion of the serratus muscle was harvested from its anterior costal insertion and used to close the pocket laterally, in order to limit implant dislocation and ensure complete coverage of the implant. The implant size was carefully chosen using trial sizers and confirmed by surgical judgment as well as by the weight of the mastectomy specimen, avoiding exerting excessive tension on the muscle and skin flap to prevent reduction of the blood supply. From 2019 onwards, indocyanine green (ICG) was routinely used to check mastectomy flap viability during surgery (Fluoptics, Cambridge, MA, USA). Clindamycin irrigation solution was used on the prosthesis before the final positioning. No sutures to recreate the IMF were used, except in two SSM patients when a lower abdominal flap was employed enhance the lower pole projection and recreate a more natural ptosis of the breast; the new inframammary fold was fixed with PDS 2-0 interrupted sutures to the thoracic fascia. The contralateral side was approached accordingly if necessary. Suction drains were placed in the subcutaneous tissue and the sub-muscular pocket, and were kept in place until the output was less than 50 cc in 24 h.

### 2.2. Statistical Analysis

All data were analyzed in terms of descriptive statistics using GraphPad Prism (version 8.0, GraphPad software, La Jolla, CA). The aesthetic scores given by the patient and the surgeon were compared using an independent parametric two-sided t-test, as values were normally distributed. Statistical significance was set at a *p*-value < 0.05.

## 3. Results

We enrolled 18 female patients in the study. The ages of patients ranged from 50 to 74 years old, with an average of 68 ± 1.8 years (AVE ± SEM). All patients presented a preoperative B (n = 7, 39%) or C (n = 11, 61%) bra cup size. In terms of comorbidities, four patients were heavy smokers, two suffered from diabetes type II, and five from hypertension. Patients underwent whole-breast external beam radiation treatment with a total radiation amount of 50 Gy (2 Gy/fraction). The interval from completion of RT to implant placement was on average 15 ± 2 years (range 2–24). Skin-sparing mastectomy was performed in five cases (28%), and nipple-sparing mastectomies in 13 (72%). Axillary dissection had already been performed at the time of BCS in seven cases (39%), while in two cases (11%) it was performed during the salvage mastectomy. The remaining nine patients (50%) underwent lymph node biopsy at the time of the last surgery. In all cases, an anatomical shape textured silicone implant was used, ranging in size from 270 cc to 350 cc (AVE ± SEM, 308 cc ± 7 cc). Breast reconstruction was combined with contralateral procedure (mastopexy, breast reduction or breast augmentation) in 10 patients (55%).

The mean hospital stay was 2 ± 0.2 days (range 2–4 days), while the time to healing (the time required to the removal of the last stich) was around 20 ± 2 days (range 14–38 days).

Adjuvant treatment after surgery included hormonotherapy in 10 patients (77%) (anastrozole, letrozole or tamoxifen) and Trastuzumab in one patient (8%).

Mean follow up was 30 months ± 2. At follow-up, all patients were free of disease. Patient data and characteristics are presented in Table 1. According to the BREAST-Q questionnaire, all patients reported high levels of “satisfaction with outcome” (which is the most important index for measuring patients’ overall sense of satisfaction) [14], with good “psychosocial wellness” and “physical impact” related to the reconstruction (see Table 2). The aesthetic evaluation showed a significant difference between the VAS scores given by the patient (mean 6.9, range 5–9) and the surgeon (mean 5.4, range 3–8) (Figure 1, Figure 2 and Figure 3).

### Complications

Evaluation of early and late complications was performed in all patients. Four patients (22%) suffered from wound dehiscence and were managed conservatively. In three patients (17%), superficial mastectomy flap necrosis required debridement and primary closure in day surgery. No implant exposure occurred in this series. Among late complications, capsular contracture was seen in seven patients (four Baker stage II and three Baker stage III, total 39%); however, no patient was willing to undergo implant exchange. In three patients (23%) a secondary lipofilling procedure was performed to enhance the aesthetic outcome and the skin tissue quality.

Complications are listed in Table 3.

## 4. Discussion

Recurrence after breast conservation therapy can be as high as 10%, and therefore previously performed surgery and radiation must be carefully considered before planning optimal future management [8]. Particularly, breast reconstruction in previously irradiated patients pose unique challenges to the reconstructive surgeon because of the detrimental effect the radiotherapy has on local tissue [9].

Autologous breast reconstruction, including pedicled and free flaps, is considered by many authors the gold standard in such cases. However, despite the favorable results expected, such procedures are generally more time-consuming and donor-site morbidity must be acknowledged. Furthermore, elderly and comorbid patients and heavy smokers are usually considered unsuitable candidates for such a complex procedure [15,16]. Concerning radiation, a higher complication rate has been described in irradiated patients (particularly, higher incidence of wound dehiscence, fat necrosis, and flap loss): intimal hyperplasia and advential fibrosis of the recipient vessels can indeed lead to vascular obstruction and anastomosis failure [17].

Implant-based breast reconstruction (IBR) today remains the most common method of post-mastectomy reconstruction [18]. However, concerns about surgical complications when performing direct-to-implant breast reconstruction have limited its application as a reconstructive option [19], and traditionally the tissue expander and implant reconstruction method is considered a more reliable technique to provide good aesthetic results and few risks [20]. However, since skin- and nipple-sparing mastectomy are considered safe options from an oncological point of view, the DTI breast reconstruction approach is gaining popularity, with DTI reconstructions accounting for 10% of implant-based breast reconstructions in 2016, according to the American Society of Plastic Surgeons. In a prospective multicenter study, Srinivasa et al. compared DTI immediate breast reconstruction and tissue-expander–implant staged breast reconstruction in more than 1400 patients with 2 years’ follow-up, and observed no differences in complication rates or patient-reported outcomes [21]. Avoiding the tissue expander, the procedure allows complete reconstruction in a shorter time, reducing patient discomfort as well as costs [22]. Moreover, there is no need for a secondary procedure, decreasing potential surgical morbidity. Patient and implant selection is of paramount importance; patients with small to moderate breast size and mild ptosis (I or II degree) are the most suitable candidates for this technique, that relies on good thickness and viability of the mastectomy flap to avoid further complications [23]. Naoum et al. analyzed in depth the effects of postmastectomy radiotherapy (PMRT) on three different breast reconstruction approaches (DTI, tissue expander and implant, and autologous), including more than 1800 breast reconstructions over a 20-year follow-up. They found that the rate of complications following DTI breast reconstruction was 50% lower than when using tissue expander and implant, and was comparable to autologous reconstruction [24]. This is probably due to the fact that DTI avoids a surgical step and the accompanying risk of inserting an implant after an expander on a previously irradiated field.

The role of IBR following radiotherapy remains extremely controversial, with earlier reports showing up to 60% complication rates with the use of tissue expander and implant reconstruction following salvage mastectomy or in previously irradiated patients [25,26,27,28,29,30]. It should be acknowledged, however, that those studies are based on low numbers and a heterogenous group of patients, including in many cases patients treated in the 1980s and 1990s with radiation protocols and devices very different from those used nowadays.

While working in a pre-irradiated field is generally a common scenario in flap surgery [31], implant-based reconstruction is more frequently adopted after RT. Persichetti et al. compared IBR (tissue expander and implant) in 20 patients with previous BCS, RT, and salvage mastectomy, versus 42 patients who underwent primary mastectomy without RT. According to their findings, no significant differences were detected in overall major or minor complication rates [27]. Recently, Cordeiro et al. reported 121 breast reconstructions using a two-stage implant-based technique in previously irradiated patients, showing a high rate of successful reconstruction with good aesthetic results over a 46-month follow-up, only slightly inferior when compared with IBR in nonirradiated breasts [12].

The use of acellular dermal matrix is debated as an adjunct in IBR. Although some evidence points to the role of ADMs in providing additional coverage for the implant, reducing the risks of implant extrusion and enhancing the aesthetic results [32], many authors have challenged this hypothesis. Chung et al. compared non-ADM and ADM reconstructive techniques in over 400 patients and found that its use was associated with a higher incidence of infection (8.9% vs. 2.1%) and seroma (14.1% vs. 2.7%), leading ultimately to a higher rate of prosthesis explantation. Thus, ADM employment in breast reconstruction should be critically assessed and possibly avoided in patients with small to medium breasts and mild ptosis [33]. It should also be considered that the use of ADM in previously irradiated fields has not been studied on a large scale and no literature on the subject can be found. However, it is intuitive to consider the previously irradiated mastectomy flap to be less than ideal for promoting the integration of the acellular membrane, potentially leading ultimately to devastating complications and reconstruction failure. In fact, prior irradiation is often considered an exclusion criterion for the use of a dermal matrix [34].

In our data series, considering the moderate average breast volume and prior irradiation in all patients, no ADMs were used.

Our data compare similarly or even favorably to previous larger scale reports on IBR after irradiated fields, in terms of mastectomy flap necrosis (13% in this series vs. 18% in the series of Cordeiro et al.) [12]. Moreover, no implant exposure occurred, albeit the small cohort examined was small. Late complications such capsular contraction reached 40% in our study, again comparable to the two-step approaches [12,27].

Regarding patient satisfaction, our results are consistent with previous research [3,14,35]. Elthair et al. administered the BREAST-Q questionnaire to 45 patients with IBR, the mean “satisfaction with breast” value was 65 points while the “satisfaction with outcome” value was 75 points [36]. Higher satisfaction rates with IBR were described when one or more fat-grafting sessions were performed before the definitive reconstruction; however, several surgical procedures were required, prolonging the reconstruction time. [35].

In this study, the high levels of satisfaction experienced by our patients may be related to two important factors: firstly, all patients had previously had breast surgery and radiotherapy, and presented a suboptimal grade of pre-mastectomy symmetry and breast shape; secondly, as stated, patients were carefully selected pre-operatively and were more concerned with surgery-related issues (such as recovery time, number of scars, and impact of surgery), hoping to return to full daily activities as soon as possible and focused less on the final aesthetic appearance.

This was reflected in the aesthetic outcome values provided, with patients generally scoring aesthetic outcomes higher than did the surgeon, consistent with previous literature where patients tended to judge the final results of reconstruction as being better if they had been through a non-reconstructed phase (e.g., delayed reconstructions) [37]. This approach should indeed be considered for proposal to older patients with reasonable aesthetic demands who seek rapid recovery and low-morbidity reconstruction. In our cohort, for instance, the secondary role played by aesthetic appearance was reflected by the fact that the great majority of patients refused even minor correction procedures (such as contour fat grafting or nipple–areola complex reconstruction).

Despite the low number of cases and the lack of a comparative cohort of patients, our data seem to confirm that the DTI strategy may be safe for use in previously irradiated fields, and should not be contraindicated a priori. However, the patient needs to be aware of the high complication rate when performing implant breast reconstruction following irradiation. Careful selection and proper education of the patient is of paramount importance; although this does not impact directly on surgical complications, it does on expectations and, by extension, on perceived outcomes. IBR following radiotherapy does not interfere with adjuvant therapies, and in cases of failure does not preclude secondary recourse to autologous reconstruction [27].

## 5. Conclusions

To the best of our knowledge, this paper represents the first report focusing on direct-to-implant breast reconstruction in previously irradiated patients.

DTI can be considered as an option for those patients who are not considered good candidates for autologous breast reconstruction. Although we acknowledge the lack of a control group receiving not DTI but standard expander–implant reconstruction, our general outcomes compare favorably with data in the literature regarding the use of staged procedures, with acceptable complication rate and patient satisfaction.

The reconstructive DTI solution best serves older patients with moderate to low aesthetic demands, seeking an all-in-one reconstructive solution and refusing an autologous procedure.

A larger cohort of patients will be necessary for statistical evaluation of outcomes and drawbacks related to the use of this technique.

## Figures and Tables

**Figure 1 jcm-11-05856-f001:**
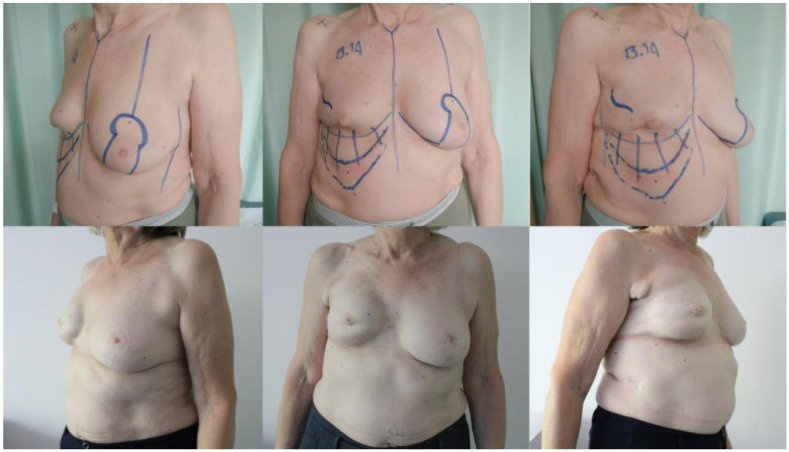
Pre-operative and post-operative pictures of a 67-year-old patient. Breast reconstruction was performed using a 335 cc anatomical implant combined with an abdominal advancement flap. The patient was satisfied with her final result, giving 8 points out of 10 according to the VAS score. Despite the occurrence of a capsular contracture, grade 2 according to Baker, the patient refused additional surgery.

**Figure 2 jcm-11-05856-f002:**
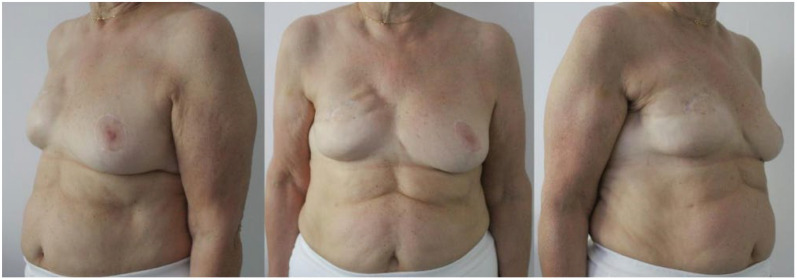
Post-operative result of a 68-year-old patient at 18-month follow-up. The right breast was reconstructed using a 320 cc anatomical implant. A contralateral reduction mammaplasty was performed in the same operative time. The aesthetic outcome reached scores of 7/10 and 6/10, according to patient and surgeon, respectively. The patient refused a fat grafting procedure to correct the medial rippling.

**Figure 3 jcm-11-05856-f003:**
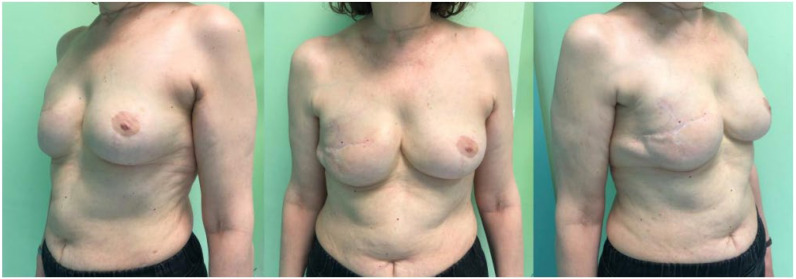
Post-operative pictures of a 62-year-old patient at 15-month follow-up. A 285 cc anatomical breast implant was used to reconstruct the right breast. In the early post-operative period, the patient presented a superficial necrosis of the upper and lower mastectomy flap, which was treated conservatively. The patient was highly satisfied with the aesthetic result (VAS 9 out of 10).

**Table 1 jcm-11-05856-t001:** Patient demographics. BCS: breast conserving surgery, GERD: gastroesophageal reflux disease, NSM: nipple-sparing mastectomy, QUART: quadrantectomy + radiotherapy, SSM: skin-sparing mastectomy.

Variable Total Patients (n = 18)	No. of Patients (%)
Mean age, years (range)	68 ± 1.8 (50–74)
Comorbidities	
Hypertension	5 (28%)
Smoking	4 (22%)
Diabetes	2 (11%)
Hypothyroidism	1 (5%)
Depression	3 (17%)
GERD	3 (17%)
Autoimmune disease	3 (17%)
Others	4 (22%)
Interval from previous radiotherapy, years (range)	15 ± 2 (2–24)
Type of mastectomy	
SSM	5 (28%)
NSM	13 (72%)
Axillary dissection	
With BCS	7 (39%)
With salvage mastectomy	2 (11%)
Type of implant	
Anatomical shape textured silicone size (range)	308 cc ± 7 cc (270–350)
Contralateral procedure	
Mastopexy	3 (17%)
Breast reduction	6 (33%)
QUART	1 (5%)
Mean hospital stay, days (range)	2 ± 0.2 (2–4)
Time to heal, days (range)	20 ± 2 (14–38)
Adjuvant treatment	
Hormonotherapy	9 (69%)
Trastuzumab	1 (8%)
Follow up, months	30 ± 2

**Table 2 jcm-11-05856-t002:** Post-operative outcomes.

Variable Total Patients (n = 18)	
BREAST-Q	
Satisfaction (breast)	62.4 (50–74)
Psychosocial wellness	65.8 (47–83)
Sexual well-being	43 (14–62)
Physical impact (chest)	69.3 (55–85)
Overall satisfaction with outcome	87.1 (76–100)
VAS	
Patients, result (range)	6.9 (5–9)
Surgeon, result (range)	5.4 (3–8)

**Table 3 jcm-11-05856-t003:** Complications after direct-to-implant reconstruction.

Variable Total Patients (n = 18)	No. of Patients
Early complications:	
Wound dehiscence	4 (22%)
Superficial mastectomy flap necrosis	3 (17%)
Late complications:	
Capsular contracture	
Baker 2	4 (22%)
Baker 3	3 (17%)
Rippling	2 (11%)
Secondary procedure	
Autologous fat grafting	3 (17%)

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
