# Peer review of "Sub-Muscular Direct-to-Implant Immediate Breast Reconstruction in Previously Irradiated Patients Avoiding the Use of ADM: A Preliminary Study"

_jcm, 2022, doi:10.3390/jcm11195856_

Round 1

Author Response

We thank the reviewer for the possibility to clarify better these concepts:

  1. In our cohort there were not parameters non normally distributed so we used a parametric test. We apologize for the typo in the manuscript. The text has been amended.
  2. We used a line with numbers on one side (from 1 to 10) for the examiner and without numbers on the other side both for patients and surgeon. The patients and the surgeon have to choose a point on the line without seeing the exact number that will be given by the examiner.

Reviewer 2 Report

The authors are to be commended on direct to implant breast reconstruction on previously irradiated patients as a preliminary study.

This article includes important results for future readers.

I have some comments below to be answered.

1.     Direct to implant breast reconstruction without ADM is challenging procedure. However, the authors successfully treated patients after radiation therapy with good results. Although study patients’ breast size was not huge, the authors need to explain their method’s advantages to put implant in small space of sub-muscular pocket.

2.     Related to my comment 1, to put implant in small space of sub-muscular pocket seems very difficult. Please add a figure of intraoperative picture which depict implant closure with muscle structures, because some future readers may doubt whether this method is possible.

Author Response

  1. We apologize if it was not clear in the manuscript the description of the surgical technique. The implant was not collocated entirely under the muscle, the superior part is covered by the pectoralis major (which maintain its attachment at the IMF but is detached from its costal insertion) and the inferior part is covered by the subcutaneous tissue of the abdominal flap that is under the inframammary fold. This is possible with the drawback of lowering the IMF of some centimeters.
  1. We thank the reviewer for the comment. Unfortunately we are not able to provide intraoperative pictures.